# Translation and Validation of the Malay Version of the Emotion Regulation Questionnaire for Children and Adolescents (ERQ-CA)

**DOI:** 10.3390/ijerph191811399

**Published:** 2022-09-10

**Authors:** Manisah Mohd Ali, Suzana Mohd Hoesni, Nur Afrina Rosharudin, Siti Rashidah Yusoff, Mohamad Omar Ihsan Razman, Khairul Farhah Khairuddin, Tuti Iryani Mohd Daud, Noor Azimah Muhammad, Dharatun Nissa Puad Mohd Kari

**Affiliations:** 1Centre for Research in Education & Community Wellbeing, Faculty of Education, Universiti Kebangsaan Malaysia, Bangi 43600, Selangor, Malaysia; 2Centre for Research in Psychology and Human Well-Being, Faculty of Social Sciences and Humanities, Universiti Kebangsaan Malaysia, Bangi 43600, Selangor, Malaysia; 3Department of Psychiatry, Faculty of Medicine, Universiti Kebangsaan Malaysia, Jalan Yaacob Latif, Bandar Tun Razak, Cheras, Kuala Lumpur 56000, Malaysia; 4Department of Family Medicine, Faculty of Medicine, Universiti Kebangsaan Malaysia, Jalan Yaacob Latif, Bandar Tun Razak, Cheras, Kuala Lumpur 56000, Malaysia; 5Department of Counselor Education and Counseling Psychology, Faculty of Educational Studies, Universiti Putra Malaysia, Serdang 43400, Selangor, Malaysia

**Keywords:** emotion regulation, adolescents, reliability, validity, Malaysian

## Abstract

The Emotion Regulation Questionnaire for Children and Adolescents (ERQ-CA) has been translated and adapted globally. This study aimed to examine the psychometric properties of the Malay version of the ERQ-CA. The ERQ-CA underwent forward and back translation twice and was tested in two separate studies, Study 1 and Study 2, with 296 and 359 students aged between 13 and 14 years old, respectively. Cronbach’s alpha values were calculated, and confirmatory factor analysis was conducted. The results from Study 1 demonstrate good internal consistency for cognitive reappraisal and expressive suppression. The results indicate a good factor loading for most of the items, but only one value of the goodness-of-fit met the criteria for a good fit. The results from Study 2 show improvements in the values of the goodness-of-fit that are comparable to previous studies, but there was a decrease in the factor loading scores. Overall, the Malay version of the ERQ-CA possesses acceptable reliability and validity. Further studies are required in the near future to develop a Malay version of the ERQ-CA that reasonably represents Malaysian adolescents.

## 1. Introduction

Experiencing emotions helps an individual to respond in a certain way to adapt to a particular environment. This capability is called emotion regulation, which refers to an individual’s ability to identify, understand, and manage their own emotions [1]. It also involves an effort to increase, decrease or maintain positive and negative emotions that may occur consciously or unconsciously [2]. This is a necessary process, where emotions can be expressed according to a situational demand or one’s goal [3].

The process model of emotion regulation by Gross and Thompson [4] proposed two distinct strategies of emotion regulation: (1) antecedent-focused and (2) response-focused strategies. Antecedent-focused emotion regulation strategies involve efforts to control and alter emotions before the emotions are elicited. An example of an antecedent-focused strategy is cognitive reappraisal (CR), where an individual changes the way he or she thinks about an event. A response-focused strategy, on the contrary, is the process of altering an emotion as the emotion is being experienced. Expressive suppression (ES) is an example of a response-focused strategy, where the individual actively inhibits their emotional expressive behavior. In short, CR is a strategy that changes the way an individual thinks about events that potentially elicit emotions, while ES is a strategy that changes the way individuals behave in response to an event.

CR and ES are believed to have different consequences for individuals. CR that occurs before an emotion is fully elicited will modify the entire emotional response, which results in fewer experiential, behavioral, and physiological responses [5]. Research has suggested that the use of CR is related to increases in positive emotions and reductions in negative emotions [5,6] and is negatively related to stress and anxiety symptoms [7]. In contrast, the ES strategy is believed to only reduce behavioral expression, yet the emotions are continually suppressed or remain unresolved [5,8]. Studies have suggested that the use of ES has a tendency to induce more negatively biased memories of negative emotions [9] and is related to depressive symptoms [10].

### 1.1. Adolescence and Emotion Regulation

Adolescence is a vulnerable stage of human development, which is characterized by significant changes in physical, cognitive, and psychosocial aspects of development. During these stages, adolescents display a lot of roles, such as being protective, accommodative, rebellious, and generous toward others [11]. This stage is also associated with frequently experiencing positive and negative emotions, instability of emotions [12], experiencing intense emotions [13] and limited access to emotion regulation strategies [14]. Given the vulnerability of emotional experiences, it is crucial that adolescents regulate their emotions effectively. The ability to regulate emotions promotes healthy development by predicting life satisfaction [15], academic achievement [16], and adolescent resiliency [17]. However, an inability to regulate emotions, such as using maladaptive strategies or less awareness of emotional states, predicts depression [10], anxiety [18], suicidal ideation [19] sleep problems in adolescence [20], and aggressive behavior [21].

Gender differences in adolescence play a significant role in emotion regulation [22,23]. In a traditional view of men and women, girls are more nurtured and accommodating, while boys are more agentic [24]. Girls are expected to express their emotions [14,25,26], in order to promote social bonding with others [27]. In contrast, boys are taught to hinder their emotions such as sadness and anxiety to portray their masculinity [25,27]. In line with studies of ER strategies, gender differences have been found specifically for ES but not for CR, in which boys were more likely to use ES than girls [28,29].

### 1.2. Emotion Regulation Measurement Tools

There are several tools that have been developed to assess emotion regulation, such as the Emotion Regulation Questionnaire (ERQ) [5], the Cognitive Emotion Regulation Questionnaire (CERQ) [30] and the Difficulties in Emotion Regulation Scale (DERS) [31]. Although originating from the same framework, the multiple and varying contents of these assessments draw attention to the inconsistencies in how emotion regulation is operationalized and limit combinations of data across studies [32]. All these assessments were mostly developed for adults and have been adapted and validated for younger people, such as children and adolescents. For example, Gullone and Taffe [33] constructed the ERQ-CA, an adaptation of the ERQ for children and adolescents, and Garnefski and colleagues [30] validated the CERQ for adolescents, while Weinberg and Klonsky [34] validated the DERS for adolescents. All these adapted versions are widely used in research on emotion regulation [35].

The DERS is a 36-item scale mainly focused on regulating negative emotional states, and it consists of six subscales (nonacceptance, goals, impulse, awareness, strategies, and clarity). According to Mazefsky et al. [36], questions regarding the DERS factor structure have been raised consistently. Some studies have argued to eliminate one of the DERS subscales, namely, awareness, leaving only five subscales [37]. Meanwhile, the CERQ is a 36-item questionnaire that focuses on an individual’s thinking process after they experienced threatening or stressful life events, and it consists of nine subscales (four subscales to assess negative emotion regulation strategies, and five subscales to assess positive emotion regulation strategies). According to Garnefski et al. [30], the CERQ merely focuses on cognitive components of emotion regulation. Lastly, the ERQ-CA is a 10-item scale that mainly evaluates responses to emotions through two subscales (CR and ES). Based on a systematic review of emotion regulation measurement [34], the ERQ-CA was found not only to consist of simplified wording and a reduction in rating scales from the original version (seven points to five points), but also to have good psychometric properties. Despite the variety of themes on emotion regulation, the ERQ-CA is one of the strongest measurements that focuses on specific strategies of emotion regulation, namely, CR and ES. This can help researchers to gain a better understanding of the essential elements of emotion regulation strategies.

### 1.3. Emotion Regulation Questionnaire for Children and Adolescents (ERQ-CA)

Since the first development of the ERQ-CA in 2012, the ERQ-CA has been widely used, translated, and adapted into countless languages such as Chinese [38], Spanish [39,40], Portuguese [41], Japanese [42], and Arabic [43]. The adaptations of the ERQ-CA have shown acceptable to good internal reliability on both the CR (ranging from 0.70 to 0.82) and ES (ranging from 0.65 to 0.71) scales [38,39,40,41,42].

In terms of validity, previous studies consistently showed that the ERQ-CA consisted of a two-factor model, comprising CR and ES, with a suitable fit [38,39,40,41,42,44]. Some studies also examined its convergent validity by exploring the relationship between depressive symptoms and CR and ES. For example, the original study of the ERQ-CA by Gullone and Taffe [33] found that CR had a weak relationship with total scores in the Children’s Depression Inventory (CDI), in a negative direction. However, there was a positive and moderate relationship between ES and total scores in the CDI. The same relationship was also found between CR and ES and depression scores when examining boys and girls aged from 13 to 15 years separately. Similarly, Liu et al. [38] examined the Chinese adaptation of the ERQ-CA and found the same direction of the CR and ES scales in relation to depression. By using the Chinese adaptation of the CDI, they found that the CR scale was negatively associated with the depression scores, while the ES scale was positively associated with the depression scores. Despite the promising potential to measure emotion regulation processes for research purposes, the ERQ-CA has not yet undergone comprehensive translation into the Malay language, the national language in Malaysia. Emotion regulation is an important component to measure among Malaysian adolescents; thus, this study will be helpful in providing a translated and validated emotion regulation tool, especially within the context of Malaysian adolescents.

In relation to gender differences in the CR and ES scales, Gullone and Taffee [33], Teixera et al. [41], and Martin-Albo et al. [40], found that there is a gender difference in the ES scale, but not in the CR scale. They found that boys scored significantly higher on ES than girls. In contrast to Gullone and Taffe [33], Pastor et al. [39] found the same CR and ES scores in boys and girls. A different approach was used by Liu et al. [38], Teixera et al. [41], Martin-Albo et al. [40], and Ng et al. [44] to examine if the factor structure of the ERQ-CA was different across genders. Using confirmatory factor analysis (CFA) by comparing models of different genders, they found that the ERQ-CA remained equivalent for boys and girls. The gender invariance in the ERQ-CA is important as differences in ERQ-CA scores primarily reflect variations in ER strategies. In previous research findings, ERQ-CA scores were not directly related to gender, and there was a response tendency between boys and girls [38]. Therefore, this research aimed to observe the structural invariance of the ERQ-CA concerning gender.

### 1.4. The Present Study

There is a major obstacle when conducting ER research in Malaysia with adolescent samples. There is no study that has yet developed or validated an instrument to measure ER strategies among Malaysian adolescents in the Malay language. Studies of ER in the Malaysian population are mainly centered on adults [45,46,47,48,49,50,51,52,53], and a Malay version of the ERQ which specifically targets adult samples has been developed [54]. While ER studies have been conducted among Malaysian adolescents [55,56,57,58], they remain scarce, and it is difficult to draw a reasonable conclusion regarding the development of Malaysian adolescents’ ER. Therefore, this study also hopes to contribute to future studies on understanding emotional regulation among adolescents in Malaysia.

Understanding ER is also necessary for Malaysians who practice collectivism [59], as some studies suggest there are differences in ER strategies between individualistic and collectivistic societies [60]. For example, previous findings have shown that collectivistic societies practice both CR and ES [61,62,63], and the ES strategy is not associated with negative outcomes such as depressive symptoms [64] and psychological functioning [65]. Therefore, it is necessary to develop the Malay version of the ERQ-CA, which was originally constructed in English, to further facilitate research in Malaysian cultures and to understand the nature of emotion regulation among collectivistic societies. For all of the reasons that have been addressed, the objective of the current study was to investigate the psychometric properties of the Malay version of the ERQ-CA among adolescents aged 13 and 14 years old. The researchers also investigated gender invariance in emotion regulation among adolescents.

Two studies were conducted to examine psychometric properties, including the internal reliability, construct validity, convergent validity, and gender differences of the ERQ-CA. The purpose of Study 2 was to enhance the psychometric properties of the Malay version of the ERQ-CA from Study 1. Through Study 1, the researchers believed that the Malay version of the ERQ-CA was not validated sufficiently and, due to the wording effects and the translation process of the items in Study 1, probably did not reflect the construct of ER strategies (or it was misunderstood by our respondents) [66]. Therefore, Study 2 was conducted to re-develop the Malay version of the ERQ-CA and re-examine its psychometric properties within the same age group, which was adolescents aged 13 and 14 years old.

## 2. Study 1

### 2.1. Materials and Methods

#### 2.1.1. Translation Process of the Malay Version of the ERQ-CA

The original ERQ-CA was translated based on the guidelines of Beaton and Bombardier [67]. The translation process involved forward and back translations in parallel by four multidisciplinary experts. All four experts in translation had qualifications such as a master’s degree in English language studies and a bachelor’s degree in education (Malay language). The forward translation (English to Malay) was conducted by two expert language translators, and a report was produced. This was followed by the back translation (Malay to English) by another two expert translators, and a report was produced. After the reconciliation of the two forward and back translations, the researchers examined the translated and original questionnaires item by item to verify their contents and chose the best translation that was closest to the original ERQ-CA. Later, two experts that specialized in psychology and education reviewed the chosen items for the final verification. Figure 1 illustrates this study’s translation and validation processes.

#### 2.1.2. Participants

Students between the ages of 13 and 14 years old (M = 13.5, SD = 0.50) voluntarily participated in this study. Thirteen- and fourteen-year-old students are in the adolescence stage where they experience the transition from primary school to secondary school in Malaysia’s education system. This transition is characterized by experiences of emotional difficulties and conflicts with parents [68], less support-seeking behavior [22], increased emotional instability and neuroticism [65], an increased rate of psychopathology [23], and it is a time of profound transformation in their emotion regulation processes [33]. The inclusion criteria for this study were participants who attended secondary government schools, were literate in the Malay language, agreed to participate, and had parental consent. Students with special needs or cognitive impairments, and those who were illiterate or who underwent psychological treatment were excluded.

In total, 296 students from three public schools, specifically in the central region of Malaysia, participated in this study. The sample consisted of 169 females (57.1%) and 127 males (42.9%). Convenience sampling was used to select the schools and the participants in the study. According to Awang [69], convenience sampling is a nonprobability sampling technique consisting of a group that is easy to reach. Only public schools registered under Malaysia’s Ministry of Education that implement co-education were included. Public schools are the mainstream choice of schools in Malaysia, which consists of several socioeconomic classes and racial ethnicities in Malaysia, such as Malay, Chinese, and Indian. The main language used in these schools is Malay.

#### 2.1.3. Procedures

Ethics approval for this study was granted by the Research Ethics Committee (REC) of the National University of Malaysia (UKM PPI/111/8/JEP-2021-182), the Ministry of Education Malaysia (KPM.600-3/2/3-eras(10243)), and the Selangor Education Department (JPNS.SPO.600-1/1/2 JLO. 14 (56)). Participants were recruited from the schools after gaining permission from the school principals and informed consent from their parents. This study was conducted during the COVID-19 pandemic, and due to the restricted movement, which affected the data collection, the collection of data was conducted online. To comply with the rules and regulations that had been set, the participants were all gathered using the Google Meet platform, with the support of trained graduate research assistants. The participants were briefed about the procedures for the survey and the duration of the study, which was approximately 20–30 min. The participants were also informed that participation was voluntary, that they could withdraw at any time without giving a reason, and that all information obtained would be treated with confidentiality. The students were then provided a link to complete questionnaires. The graduate research assistants accompanied all participants throughout the sessions to entertain any inquiries from the participants regarding the items in the questionnaires and to ensure all the participants submitted their responses. Lastly, all online questionnaires completed by the participants were gathered for further analysis.

#### 2.1.4. Measures

*The Malay version of the Emotion Regulation Questionnaire for Children and Adolescents (ERQ-CA)*. The original ERQ-CA by Gullone and Taffe [33] consists of 10 items and is a revised version of the Emotion Regulation Questionnaire (ERQ) [5]. Permission was obtained from the original authors to translate and use the questionnaire in this study. The Malay version of the ERQ-CA comprises a two-factor structure, with CR (six items), and ES (four items). The CR strategy involves reframing a potentially emotion-eliciting event in such a way that its emotional impact is altered. Meanwhile, the ES strategy involves the inhibition of continuing emotional-expressive behavior. The Malay version of the ERQ-CA uses a 5-point Likert scale ranging from 1 (strongly disagree) to 5 (strongly agree). The scale scores were calculated based on the participants’ total item scores on each scale.

*The Malay version of the Depression, Anxiety, and Stress Scale-21 (DASS-21)*. The original DASS-21 was developed by Lovibond and Lovibond [70] and has been translated into the Malay language by Musa, Fadzil, and Zain [71]. It is an open-access survey and consists of 21 items that are used to measure recent emotional states relating to stress, anxiety, and depression. The Depression scale assesses dysphoria, hopelessness, devaluation of life, self-deprecation, lack of interest/involvement, anhedonia, and inertia. The Anxiety scale evaluates autonomic arousal, skeletal muscle effects, situational anxiety, and subjective anxious affect experiences. The Stress scale measures persistent non-specific arousal levels. It evaluates restlessness, anxious arousal, being easily upset/agitated or irritable/over-reactive and impatient psychological aspects. Each subscale encompasses seven items that are rated on a 4-point Likert scale. The scale ranges from 0 (did not apply to me) to 3 (applied to me very much), with higher scores indicating greater symptomatology. In general, the DASS was selected because it is suitable for non-clinical samples and for screening of adolescents and adults [72]. At the same time, Lovibond [69] also suggested that the DASS-21 is more suitable for research purposes. For the purpose of examining the convergent validity of the Malay version of the ERQ-CA, we only used the Depression and Anxiety scales, with a total of 14 items. The chosen scales were based on previous studies that found that the adaptation of the ERQ-CA was correlated with depression [33,38] and anxiety [18].

#### 2.1.5. Statistical Analyses

The data were analyzed to determine the reliability and validity of the Malay version of the ERQ-CA. Cronbach’s α values for CR, ES, and all items were calculated to test the reliability of the Malay version of the ERQ-CA. Confirmatory factor analysis (CFA) with maximum likelihood estimation was conducted to evaluate the validity of the Malay version of the ERQ-CA. The chi-square statistic (*x*^2^), relative chi-square (*x*^2^/*df*), goodness-of-fit index (GFI), comparative fit index (CFI), and root mean square error of approximation (RMSEA) were used to determine the factor structure of the Malay version of the ERQ-CA. The value of the *x*^2^(*df*) should be non-significant and *x*^2^/*df* should be below 0.50 to imply a good fit of the model [65]. Values of the GFI and CFI imply an acceptable fit if they are more than 0.09. Values that are less than 0.08 indicate a moderate fit for the RMSEA, and values that are higher than 0.08 indicate an unacceptable fit [73]. A summary of the omnibus test and indices of a model with a good factor structure is shown in Table 1. The factor loading of each item was evaluated. Values that are higher than 0.5 indicate that the item can measure the respective construct, i.e., CR or ES [69,74]. Correlation analysis (i.e., Pearson’s *r*) was also conducted between the scales of the Malay version of the ERQ-CA and the Depression and Anxiety scales of the DASS-21 to examine convergent validity. An independent sample *t*-test was also conducted to examine gender differences for each of the scales in the Malay version of the ERQ-CA. IBM SPSS Statistics version 26 (Armonk, NY, USA) was used to conduct reliability analysis and correlational analysis. IBM SPSS Amos version 24 (Chicago, IL, USA) was used to analyze the factor structure.

### 2.2. Results and Discussion

#### 2.2.1. Descriptive Statistics

The majority ethnic group of this study was Malays (82.8%), followed by Indians (13.5%), Chinese (2%), Bumiputeras (1%), and others (0.7%). Table 2 shows the means and standard deviations for CR, ES, and each item in the Malay version of the ERQ-CA.

#### 2.2.2. Internal Consistency Reliability

Descriptive data of the Malay version of the ERQ-CA included means, standard deviations, skewness, kurtosis, and corrected item-total *r* and can be seen in Table 2. The overall Cronbach’s α value for the Malay version of the ERQ-CA was 0.61. Cronbach’s α value was 0.67 for the CR scale and 0.68 for the ES scale. Only items 1, 7, 8, 9, and 10 had a corrected item-total *r* of slightly more than 0.30, while items 2, 3, 4, 5, and 6 had a corrected item-total *r* of less than 0.30.

These findings indicate that the reliability of the Malay version of the ERQ-CA is acceptable. The alpha value results are consistent with those reported by Pastor et al. [39] for their Spanish translation of the ERQ-CA. Furthermore, the results are partly consistent with the Spanish [40] and Portuguese [41] translations. However, the overall alphas in our study are lower than those reported in the original version [33], the Chinese adaptation [38], and the Japanese adaptation of the ERQ-CA [42]. In addition, items 2, 3, 4, 5, and 6 did not correlate well with the overall scale and, as suggested by Field [75], should be dropped. However, based on previous studies on the reliability of the ERQ-CA, none of the items were removed from the instruments [33,38,39]. Gong et al. [76] also suggested that deleting a few items in the ERQ-CA may lead to an imbalance in measuring emotion regulation constructs and that it neglects the main components of emotion regulation.

#### 2.2.3. Construct Validity

The results of the goodness-of-fit indicator were *x*^2^(34) = 72.76, *p* < 0.01, *x*^2^/*df* = 2.14, GFI = 0.863, CFI = 0.781, and RMSEA = 0.116. Only one value of the indices met the cut-off fit for a model with a good factor structure. The standardized item factor loadings ranged from 0.20 to 0.8 for CR and from 0.51 to 0.67 for ES. Specifically, the factor loadings for all items were greater than 0.50, except for item 1, with a factor loading of 0.45 and item 5, with a factor loading of 0.20.

Following the recommendation of the original study by Gullone and Taffe [33] and other adaptation studies [41,42,44], a correlated error was added between item 1 and item 3 to produce a better fit. Nonetheless, the model still had a poor fit, with the following values: *x*^2^(33) = 72.12, *p* < 0.01, *x*^2^/*df* = 2.185, GFI = 0.864, CFI = 0.779, and RMSEA = 0.118. The standardized item factor loadings were not improved such that CR items ranged from 0.20 to 0.78 and ES items ranged from 0.51 to 0.65. In addition, the factor loadings of items 1 and 5 remained below 0.50. There was also no correlation between CR and ES, *r* = 0.18, *p* = 0.25.

According to Awang [77] and Chua [74], when an item has a factor loading of less than 0.5, this indicates that the item is less likely to measure the unobserved variable. Therefore, items 1 and 5 in Study 1 were less likely to measure the CR strategy. Furthermore, the results of the CFA showed a poor fit of the two-factor structure of the Malay version of the ERQ-CA. The poor fit of the model could be explained by the wording effects which could affect adolescents’ comprehension and judgment of the Malay version of the ERQ-CA items [66]. Thus, the translated items might not reflect the respective ER strategies as we expected or were misunderstood by the adolescents.

#### 2.2.4. Convergent Validity

Correlations were calculated between the two scales of the Malay version of the ERQ-CA and the Depression and Anxiety scales of the DASS-21. A significant negative relationship was observed between depression total scores and the CR scale, *r* = −0.22, *p* < 0.01. Furthermore, there was a significant positive relationship between depression total scores and the ES scale, *r* = 0.53, *p* < 0.01. The Anxiety scale was also observed to be significantly correlated with both the CR (*r* = −0.12, *p* < 0.05) and ES scales (*r* = 0.51, *p* < 0.01), with negative and positive relationships, respectively. A summary of the results can be seen in Table 3. As expected, the Malay version of the ERQ-CA portrayed sound convergent validity, which is consistent with previous adaptation studies [33,38]. There was a negative relationship between the scores of the Depression and Anxiety scales and the CR scale scores. Furthermore, this study also found a positive relationship between the Depression and Anxiety scores and the ES scale.

#### 2.2.5. Gender Difference

An independent sample *t*-test showed that there was no significant difference in CR scores (*t*(295) = 0.311, *p* = 0.76, two-tailed) for boys and girls. A summary of the results can be seen in Table 4. The magnitude of the differences in the means (mean difference = 0.121, 95% CI [−0.65, 0.89]) was very small (eta squared = 0.003). However, the significant gender difference in ES scores (*t*(295) = −2.31, *p* < 0.05, two-tailed) indicates that the girls use the ES strategy more than the boys. The magnitude of the differences in the means (mean difference = −0.85, 95% CI [−1.58, −0.13]) was small (eta squared = 0.02). In relation to gender, the findings show that there was no gender difference in CR scores, and this is consistent with the findings of the original study’s ERQ-CA [33], the Spanish adaptation [39,40], and the Portuguese adaptation [41]. Meanwhile, girls reported using ES more often than boys. This is inconsistent with Gullone and Taffe [33] and Martin-Albo et al. [40], who found that boys used ES more compared to girls, as well as Pastor et al. [39], who did not find gender differences in both CR and ES.

## 3. Study 2

### 3.1. Methods

#### 3.1.1. Participants

This study included a total of 359 students recruited from public schools in the central and northern regions of Malaysia. Convenience sampling was used to select the schools and participants in the study. Study 2 used the same inclusion and exclusion criteria as Study 1. A total of 202 females (56.3%) and 157 males (43.7%) aged between 13 and 14 years old (SD = 0.50) participated in the study. The majority of the participants were Malays (79.4%), followed by Chinese (8.6%), Indians (6.4%), Bumiputeras (4.5%), and others (1.1%).

#### 3.1.2. Translation Process of the Malay Version of the ERQ-CA

The augmentation of the Malay version of the ERQ-CA followed on from Study 1. It went through another translation process based on the guidelines of Beaton and Bombardier [67]. The translation process underwent the same process as Study 1 but with another four language experts: two with a bachelor’s degree in Education (Honors) Teaching English as a Second Language), and two with bachelors degree in Education (Malay Language). The first two language experts were involved in the forward translation (English to Malay), while the other two language experts were involved in the back translation (Malay to English). Both versions were compared, and a reconciliation of this translation was reached to verify the contents and choose the best translation that was close to the original ERQ-CA. Three experts specialized in psychology and education reviewed the chosen items for the final verification. The final version of the Malay version of the ERQ-CA then underwent refinement by a Malay language expert (bachelor’s degree in education (Malay language)) to strengthen the validity of the instrument. The Malay version of the ERQ-CA was also pre-tested among three students to assess the clarity of the Malay version of the ERQ-CA [78]. Figure 2 presents the final translation of the ERQ-CA.

#### 3.1.3. Procedures

Ethics approval for this study was granted by the Research Ethics Committee (REC) of the National University of Malaysia (UKM PPI/111/8/JEP-2021-182), the Ministry of Education Malaysia (KPM.600-3/2/3-eras(10243)), the Selangor Education Department (JPNS.SPO.600-1/1/2 JLO. 14 (56)), the Kuala Lumpur Education Department (JWPKL.600-9/1/5 Jld.3(12)), and the Perlis Education Department (JPNPs.SP.600-1/1/1 JLD. 4 (07)). After obtaining approval from the school principal, participants from three different schools were selected. All participants gave informed consent to participate and obtained permission from their parents. Similar to Study 1, the data collection was conducted online, and the DASS-21 and the newly translated Malay version of the ERQ-CA were administered to the participants.

#### 3.1.4. Statistical Analyses

The reliability analysis, criteria for model fit, correlation analysis, and software program used were the same as those used in Study 1. An additional CFA was run on the model to examine structural invariance across genders [38,40,41,44] if the model fit on the total sample was observed. There is gender invariance if the difference in the CFI (ΔCFI) is ≤0.010 and the difference in the RMSEA (ΔRMSEA) is ≤0.015 between the ERQ-CA models for boys and girls [79].

### 3.2. Results and Discussion

#### 3.2.1. Descriptive Statistics

The mean and standard deviation for each scale and item of the Malay version of the ERQ-CA are shown in Table 5.

#### 3.2.2. Internal Consistency Reliability

Table 5 shows a summary of the mean, standard deviation, skewness, kurtosis, and corrected item-total *r*. The overall Cronbach’s α value for the Malay version of the ERQ-CA in Study 2 was 0.70. Meanwhile, Cronbach’s α value was 0.72 for the CR scale and 0.58 for the ES scale. The corrected item-total *r* for all the items was more than 0.30.

The results show that the new translated Malay version of the ERQ-CA is moderately reliable. Our result for CR is similar to that of the Portuguese adaptation [41] and Chinese adaptation [38] but lower than that of the original study [33] and the Spanish adaptation [40]. Meanwhile, the result for ES is similar to that of Martin-Albo [40], but lower than that of Gullone and Taffe [33], Liu et al. [38], Teixera et al. [42], and Pastor et al. [39].

#### 3.2.3. Construct Validity

The results of the goodness-of-fit indicator were *x*^2^(*df*) = 120.95, *p* < 0.01, *x*^2^/*df* = 3.56, GFI = 0.94, CFI = 0.86, and RMSEA = 0.09. Two values of the indices met the cut-off fit for a model with a good factor structure. The standardized item factor loadings ranged from 0.37 to 0.69 for CR and from 0.34 to 0.72 for ES. Specifically, all the items had a standardized factor loading of more than 0.50, except for items 1, 2, 4, and 5.

By adding a correlation between the errors for item 1 and item 3 [33,41,42], the fit of the model was improved. Three indices met the cut-off fit for a model with a good factor structure, with overall values of *x*^2^(*df*) = 112.33, *p* < 0.01, *x^2^*/*df* = 3.40, GFI = 0.94, CFI = 0.87, and RMSEA = 0.080. The standardized item factor loadings ranged from 0.34 to 0.71 for CR and from 0.33 to 0.72 for ES. Specifically, all the items had a standardized factor loading of more than 0.50, except for items 1, 2, 3, 4, and 5. A summary of the indices is shown in Table 6. In addition, there was also a significant correlation between CR and ES, *r* = 0.48, *p* < 0.50.

In general, the results indicate that the translated ERQ-CA in Study 2 has a two-factor structure, similar to previous studies [38,39,40,41,42,44]. Even though some of the items had factor loadings below 0.50, and an additional item was found to load below 0.5 after correlating between the errors, the model achieved a suitable fit, which is comparable to Liu et al. [38], Teixera et al. [41], Martin-Albo et al. [40], and Ng et al. [44]. For this reason, we decided not to drop any items from the model, which would affect the measured construct of ER [67].

#### 3.2.4. Convergent Validity

A significant positive relationship was only observed between Depression total scores and CR total scores, *r* = −0.11, *p* < 0.05. No significant relationship was found between CR and Anxiety scores (*r* = 0.06, *p* = 0.26), ES and Depression scores (*r* = 0.08, *p* = 0.16) or ES and Anxiety scores (*r* = 0.70, *p* = 0.15). Table 7 presents a summary of the results. Different from our expectations and from previous studies [7,31,37], the results of the correlation between CR and ES show different directions. It was found that only CR was correlated with the Depression scale, but in a positive direction. Moreover, we expected ES to correlate with the Depression and Anxiety scales [33,38], but non-significant results were observed. It was suggested by Aldao [80] that any ER strategy could be adaptive or maladaptive depending on the individual and social context. In our sample, the ES strategy might not have been treated as a maladaptive strategy, which could potentially affect the adolescents’ well-being and mental health.

#### 3.2.5. Gender Differences

An independent sample *t*-test showed that there was no significant difference in CR scores (*t*(364) = −0.47, *p* = 0.64, two-tailed) for boys and girls. The magnitude of the differences in the means (mean difference = 0.02, 95% CI [−0.77, 0.723]) was very small (eta squared = 0.00). Similarly, no significant difference in ES scores (*t*(295) = −1.69, *p* = 0.09, two-tailed) was found for boys and girls. The magnitude of the differences in the means (mean difference = −0.85, 95% CI [−1.07, 0.08]) was very small (eta squared = 0.007). The means and standard deviations of CR and ES between boys and girls can be seen in Table 8.

We ran an additional CFA on the model to examine if the ERQ-CA structure was invariant across genders [33,38,40,41,44]. The fit for boys and girls was slightly better after adding a correlation between the errors for item 1 and item 3. The values of the model indices for boys were *x*^2^(23) = 74.26, *p* < 0.01, *x*^2^/*df* = 2.25, GFI = 0.91, CFI = 0.85, and RMSEA = 0.07, which were better than those for girls, with *x*^2^(23) = 105.52, *p* < 0.01, *x*^2^/*df* = 3.20, GFI = 0.91, CFI = 0.80, and RMSEA = 0.10. The model for boys (three indices met the criteria for a good model fit) was better than the model for girls (two indices met the criteria for a good model fit). Furthermore, the result of the model invariance met the criterion for the RMSEA (ΔRMSEA = −0.03), but not for the CFI (ΔCFI = 0.056). A summary of the indices can be seen in Table 6.

Consistent with the original study [33] and the Spanish adaptation [39], the result shows that use of the CR strategy did not differ between boys and girls. For ES, however, we found that use of the ES strategy did not differ between boys and girls, which is similar to the study by Pastor et al. [39], but not the original study [33] or the Portuguese adaptation [41]. It might be possible that the adolescent boys and girls in this study tend to suppress their emotions in a similar way, but which is different from the way Australian [33] and Portuguese adolescents suppress their emotions [41]. In terms of the ERQ-CA as a whole, the model invariance between boys and girls was not clearly demonstrated. This finding is different from that of the original study [33], the Chinese adaptation [38], the Spanish adaptation [39,40], and the Portuguese adaptation of the ERQ-CA [41]. This could be due to girls being more aware of their emotions and more engaged with their emotions as they grow older. This would manifest in their lives via a higher likelihood of them experiencing emotional problems and anxiety-depressive symptoms in the adolescent stage [14].

## 4. General Discussion

This study aimed to provide information on the psychometric evaluation of the Malay version of the ERQ-CA. The Malay version of the ERQ-CA was administered among adolescents aged 13 and 14 years old who went to public schools in Malaysia. The translation was carried out using forward and backward procedures in both studies, except that in Study 2, a Malay teacher was recruited to refine the items, and three adolescents were recruited to assess the clarity and understanding of each item of the Malay version of the ERQ-CA.

Overall, the psychometric properties of the Malay version of the ERQ-CA were improved after Study 2. The results from Study 2 show that the factor structure of the Malay version of the ERQ-CA was the same as that of the original version [33] and versions from other adaptation studies [38,39,40,41,42,44]. Although the alpha value of ES was lower compared to Study 1, it was still within the acceptable range of reliability. We believe that the low alpha values may be due to cultural peculiarities, the age range of the sample, or the small sample size. Tavakol and Dennick [81] also suggested another possible reason; that is, a smaller number of items in the questionnaire might influence the alpha results.

In regard to the convergent validity, Study 1 portrayed the convergent validity of the Malay version of the ERQ-CA that we expected, but the results of Study 2 were different from our expectations. In Study 1, we found that CR and ES were related to the Depression and Anxiety scales, but we found that only CR was correlated with the Depression scale in Study 2. Similarly, to the gender differences in CR and ES, we found that CR did not differ across genders in both studies as expected. However, we found results in ES differed from our expectations. Specifically, the results from Study 1 demonstrate that girls were more likely to use ES than boys. However, there was no gender difference in ES in Study 2. For the gender difference in the overall ERQ-CA, we found that boys regulate their emotions more effectively than girls. The finding of Domes et al. [82] suggest that men’s brain activity in emotion regulation processing areas is more effective compared to women. Therefore, further investigation needs to be carried out to replicate these findings.

## 5. Limitations and Future Research

The current study demonstrated that the Malay version of the ERQ-CA possesses acceptable reliability and construct validity. Further studies need to be conducted to replicate the results of the convergent validity and gender difference. Furthermore, the sample used in this study was not representative of all Malaysian adolescents as Malaysia is a country with a diverse culture [58]. Extended research is crucial, which can be carried out by increasing the number of participants and conducting a stratified sampling technique to address the diversity issue, which is important for verifying the psychometric properties of the Malay version of the ERQ-CA. Furthermore, further analysis of CFA should be conducted to compare urban vs. suburban population or across developmental stages (e.g., early vs. middle vs. late adolescence) to ensure the Malay version of the ERQ-CA is a valid tool to measure emotion regulation strategies among Malaysian adolescents regardless of demographic backgrounds.

## 6. Conclusions

In conclusion, the current study provided the initial results of the psychometric properties of the Malay version of the ERQ-CA using adolescents who reside in Selangor, Wilayah Persekutuan Kuala Lumpur and Perlis. The results show that the Malay version of the ERQ-CA is reliable, but improvements should be made for it to be considered a valid instrument to assess emotion regulation strategies among Malaysian adolescents. We suggest that further comprehensive studies that appropriately represent Malaysian adolescents should be conducted to obtain better psychometric properties of the Malay version of the ERQ-CA.

## Figures and Tables

**Figure 1 ijerph-19-11399-f001:**
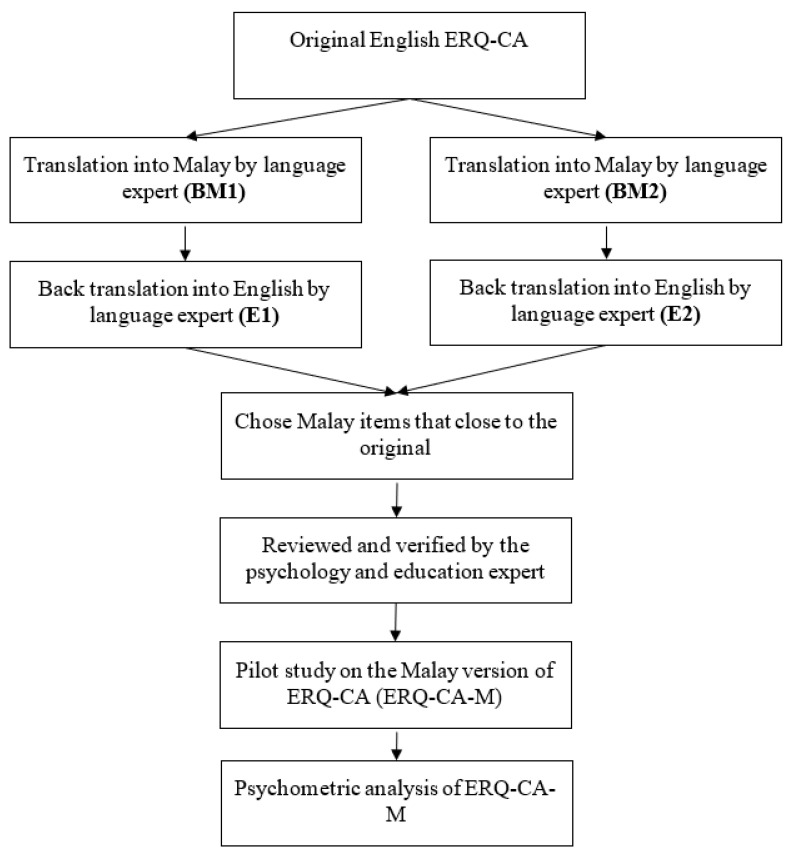
Overview of the translation and validation processes of the Malay version of the ERQ-CA.

**Figure 2 ijerph-19-11399-f002:**
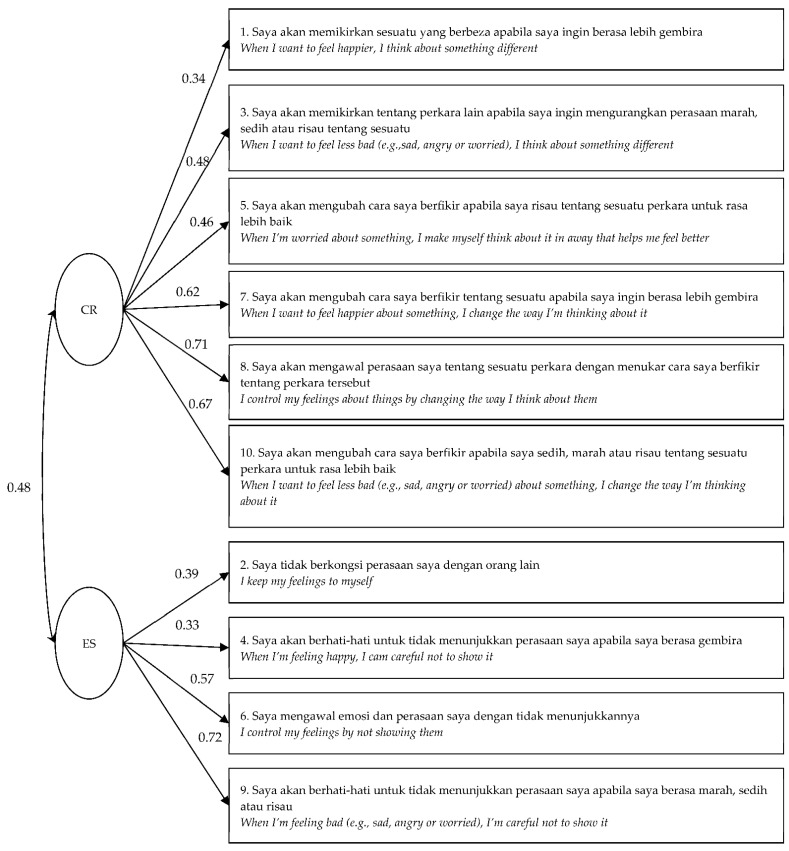
The Malay translation and the confirmatory factor analysis model of the Malay version of the ERQ-CA. The paths indicate the standardized regression coefficients for the 13- and 14-year-old sample.

**Table 1 ijerph-19-11399-t001:** Goodness-of-fit assessment.

Omnibus Test	Criterion for Goodness-of-Fit
$x^{2}$(*df*)	Non-significant
$x^{2}$/*df*	≤0.50
GFI	≥0.90
CFI	≥0.90
RMSEA	≤0.08

**Table 2 ijerph-19-11399-t002:** Mean (M), standard deviation (SD), skewness, kurtosis, and corrected item-total *r* for all items of the Malay version of the ERQ-CA in Study 1.

Scales/Items	M	SD	Skewness	Kurtosis	Corrected Item-Total *r*
CR					
Overall CR	22.01	3.33			
1	3.68	0.86	−0.71	0.59	0.34
3	3.86	0.89	−0.79	0.55	0.28
5	3.66	0.97	−0.72	0.19	0.05
7	3.56	0.88	−0.72	0.54	0.35
8	3.62	0.90	−0.65	0.28	0.32
10	3.63	0.87	−0.66	0.43	0.39
ES					
Overall ES	13.43	3.18			
2	3.37	1.25	−0.33	−1.02	0.26
4	2.98	1.20	0.05	−0.99	0.24
6	3.38	0.99	−0.19	−0.6	0.28
9	3.69	1.03	−0.66	−0.15	0.36

CR = cognitive reappraisal; ES = expressive suppression.

**Table 3 ijerph-19-11399-t003:** Correlations between the Malay version of the ERQ-CA and the DASS-21 scales.

Scales	Depression	Anxiety
CR	−0.22 **	−0.12 *
ES	0.53 **	0.51 **

CR = cognitive reappraisal; ES = expressive suppression. * *p* < 0.05; ** *p* < 0.01.

**Table 4 ijerph-19-11399-t004:** Mean (M) and standard deviation (SD) of CR and ES in boys and girls.

Sample	CR		ES	
	M	SD	M	SD
Boys	22.09	3.04	12.95	2.86
Girls	21.96	3.53	13.80	3.35

**Table 5 ijerph-19-11399-t005:** Mean (M), standard deviation (SD), skewness, kurtosis, and corrected item-total *r* for all items of the Malay version of the ERQ-CA in Study 2.

Scales/Items	M	SD	Skewness	Kurtosis	Corrected Item-Total *r*
CR					
Overall CR	23.95	3.03			
1	4.04	0.67	−1.56	5.61	0.33
3	4.11	0.82	−1.43	2.91	0.43
5	3.94	0.84	−1.48	2.93	0.37
7	3.96	0.77	−1.73	4.62	0.55
8	3.88	0.79	−1.49	2.84	0.53
10	4.01	0.78	−1.57	3.86	0.53
ES					
Overall ES	14.64	2.78			
2	3.48	1.16	−0.66	−0.66	0.38
4	3.42	1.11	−0.73	−0.65	0.32
6	3.87	0.94	−1.51	2.26	0.30
9	3.88	0.96	−1.334	1.64	0.45

CR = cognitive reappraisal; ES = expressive suppression.

**Table 6 ijerph-19-11399-t006:** Comparison of confirmatory factor analysis (CFA) models in Study 2.

	CFA Model											
	Simple					e1 and e3 Correlate					ΔCFI	ΔRMSEA
Group	$x^{2}$ (df)	$x^{2}$/df	GFI	CFI	RMSEA	$x^{2}$ (df)	$x^{2}$/df	GFI	CFI	RMSEA		
Total sample	120.59 (34)	3.56	0.94	0.86	0.09	112.13 (33)	3.40	0.94	0.87	0.08		
Gender												
Boys	74.28 (34)	2.19	0.91	0.86	0.09	74.26 (33)	2.25	0.91	0.86	0.07		
Girls	122.12 (34)	3.59	0.89	0.76	0.11	105.52 (33)	3.19	0.91	0.80	0.10	0.06	−0.03

**Table 7 ijerph-19-11399-t007:** Correlations between scales of the Malay version of the ERQ-CA and the DASS-21.

Scales	Depression	Anxiety
CR	0.11 *	0.06
ES	0.08	0.70

CR = cognitive reappraisal; ES = expressive suppression. * *p* < 0.05.

**Table 8 ijerph-19-11399-t008:** Mean (M) and standard deviation (SD) of CR and ES in boys and girls.

Sample	CR		ES	
	M	SD	M	SD
Boys	23.86	2.95	14.33	2.73
Girls	24.01	3.10	14.88	2.82

## Data Availability

The data is available upon request from the authors.

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
