# Peer review of "Translation and Validation of the Malay Version of the Emotion Regulation Questionnaire for Children and Adolescents (ERQ-CA)"

_ijerph, 2022, doi:10.3390/ijerph191811399_

Round 1

Reviewer 1 Report

In this manuscript, the psychometric properties of the Malay version ERQ-CA-M are examined. While this study can be a first approach to the validation of this instrument, I have some concerns about the relevance of the content. I find especially problematic that the findings do not support the validity of this questionnaire to be used. In my opinion, the authors should firstly advance in the development of the instrument to reach an acceptable model fit. Secondly, I wonder whether a preliminary study on psychometric properties is significant enough to contribute to the existing literature in the field. I think the work would be more significant if some other findings (aside from the psychometric properties of the scale) were reported when publishing the validation of the scale. Some comments are provided for the specific sections of the work. I do hope these comments can help the authors to improve their work.

Introduction

In my opinion, the authors nicely present the topic and the construct under study. They also develop on the ERQ-CA. It would be interesting, as a link between sections 1.1 and 1.2, to provide some more information about how the constructs under study have been previously assessed. Which tools have been used? What are their strengths and weaknesses? Why did authors choose the ERQ?

 At the end of both sections 1.1 and 1.2 the authors talk about gender differences, which aligns with part of the presented findings. I think a stronger rationale should be provided to address gender differences.

It is not clear to me why the authors mention emotional intelligence in the first paragraph of the present study section.

Method

The authors should mention on what basis they chose the translation procedure or the sampling method. Furthermore, some more information on ethical issues should be provided. In this regard, some references on methodological and ethical issues are missing.

The data analysis and results section should be aligned. The corrected-item total r test do not currently appear in the data analysis section.

Given that the aim of the study is the validation of an instrument, I think the items should be explicitly presented.

Results

This is in my opinion the weakest section in the manuscript. The findings do not support the validity of this instrument to be used in future studies as it is. As the authors indicate, the goodness-of-fit indices did not meet the widely accepted criteria. Furthermore, the analyses display important problems regarding the items suitability for the scale.

Corrected-item total r for item 5 (.05) do not seem acceptable. A correlation value less than 0.2 or 0.3 indicates that the corresponding item does not correlate very well with the scale overall and therefore should be dropped (Everitt, 2000; Field, 2005)

On the other hand, the authors mention that two items do not reach the acceptable threshold for factor loading. While the authors establish 0.5 as an acceptable cut-off value, according to Awang (2014), while this value could be used for completely new scales, for a previously developed scale, the factor loading for every item should be 0.6 or higher (Awang, 2014). Any item having a factor loading less than 0.6 and an R2 less than 0.4 should be deleted from the measurement model. Although the authors do mention this issue in the discussion, I do not think they provide some convincing reasons to maintain the items.

In my opinion, these problems might be affecting the model of fit indices and should be addressed before sharing this work with the scientific community.  

Other minor comments:

Just before Table 1 they mention that M and SD for dimensions are presented but they are not.

In Figure 2, the authors mention “the two confirmatory factor analysis” but it is only one.

Awang, Z. (2014). Research methodology and data analysis (2nd ed.). Universiti Teknologi Mara, Malaysia: UiTM Press

Everitt, B.S. (2002) The Cambridge Dictionary of Statistics, 2nd Edition, CUP. 

Field, A., (2005). Discovering Statistics Using SPSS. 2nd ed. London: Sage

Author Response

Dear Reviewer, 

Thank you so much for taking the time to review and give valuable comments to our manuscript. We have replied to your comments and questions in the attached Word Document. Looking forward to your feedback. 

Regards

Reviewer 2 Report

Introduction part.

Authors should diligently check and arrange the references so that the spelling of the surnames matches the surnames in the References part, as well as the entry of surnames in all references should be the same if it is the same article and author. They cannot mention only one last name in the reference if there are several authors of the article or to write only three of the four authors. The publication year of the article must also match both in the reference and in the References part. Of course, all the sources mentioned in the references should be presented in the Reference part, but many authors are missing.

Methods

The authors have not indicated whether the ERQ-CA survey had an open access or whether the authors had to obtain permission from the authors of the original survey. It is not clear why the authors choose the age 13-14? This should be explained why such a narrow age range was chosen. I believe that the authors cannot write "Emotion Regulation Questionnaire for Malaysian Adolescents (ERQ-CA-M)" and use such an abbreviation if it is a Preliminary Study, the survey is not standardized for Malaysian teenagers, only the ERQ-CA is adapted for a narrow age range. It is not clear whether a pilot study was conducted after the translation to test the translated test before the adolescents in this study were asked to complete it? It has not been described. The impression remains that this is a pilot study to test the translation.

Results.

The authors present Cronbach's alpha for both scales, but not for the survey as a whole and for each item separately, nor do they compare it with the original survey. Discrimination index for each item, discrimination index within survey boundaries and factor boundaries has not been calculated either. No interscale correlation was calculated, but the authors calculated the correlation with another instrument.

Discussion.

It would be better to compare the difference in results on the ES scale between genders with the results of other adaptation studies.

References.

The year of publication must be indicated for all sources, there is no need for full authors' first names, but only initials. All sources cited here should also appear in the text.

Author Response

(The authors gave the same response as above.)

Reviewer 3 Report

Peer review report for the manuscript “Malay Validation of the Emotion Regulation Questionnaire for Children and Adolescents (ERQ-CA): A Preliminary Study.”

In this manuscript, the authors describe a cross-sectional study that evaluated the psychometric properties of the Malay translation of the Emotion Regulation Questionnaire for Children and Adolescents (ERQ-CA), developed by Gross and John (2003). The study found that the Malay version of ERQ-CA is reliable but not sufficiently portrayed as a valid instrument to assess adolescent emotion regulation strategies.

The manuscript is well written and organized. The authors clearly explained the complexity of their study approach, methodology, measures, data analysis, and findings step by step. Mental health is a primary concern worldwide, especially in the post-COVID lockdown era. The study is deemed relevant for the journal readers and worthy of publication. I have some comments and suggestions.

1.      In sub-section 1.2.,

a.       May the authors clarify if Gross and John (2003) examined a sample of undergraduate students?

b.      I suggest using the word “supported” to replace “revealed” In the sentence “The confirmatory analysis (CFA) also revealed ERQ has two-factor structure.”

c.       I suggest the authors mention the instrument’s original language. I assume it is English since the study was conducted in Australia. If so, I suggest the authors indicate the language translated in the US.

2.      In sub-section 2.2, I suggest the authors rearrange the two paragraphs, using the second one as the opening for the sub-section.

3.      In sub-section 2.4, I suggest the authors briefly explain the relevance of the Malay version of Depression, Anxiety, Scale and Stress-21 (DASS-21) in their study.

4.      In sub-section 3.1,

a.       the authors mentioned, “Each value of the indices did not meet the cut-off fit of a good factor structure model.” What is the cut-off fit of a good factor structure model?

b.      I do not understand this sentence “However, the factor loadings for all items were greater than 0.50 except item 1 and 5 which the CR construct can be explained by 45% and 20% respectively.” I suggest re-wording.  

5.      In the first paragraph of the Discussion section, I suggest the authors mention that the participants were students in public school for consistency with the methods section.

6.      In the last paragraph of the Discussion Section, I suggest the authors provide the rationale for not dropping the items as Field (2017) recommended.

Thank you for a read-worthy manuscript.

Author Response

(The authors gave the same response as above.)

Round 2

Reviewer 1 Report

I would like to congratulate the authors for the excellent work done on the manuscript.

I think they have thoroughly addressed my comments. It is especially relevant that they have augmented the sample size to revise the psychometric properties in a second study. I would only want to make two minor comments to the current version of the manuscript.

The first one is that is still not clear for me why the authors explicitly mention emotional intelligence in the last part of the introduction (page 5 line 25-27) to show the relevance of studying emotion regulation. I recommend the authors either to delete this information or to revise the first part of the introduction to make sure that the relevance of addressing the study of emotional intelligence and the contribution that the development of this tool would make to this aim is clear.

Secondly, to keep consistency thorough the manuscript, I would not include a subsection named “The present study” before the Methods for the Second Study. In my opinion it would make more sense to explain the entire process in an only “Present Study” section and then, develop the Method and Results sections for each of both studies as it is now.

Author Response

Dear Reviewer, 

Thank you for your constructive feedback. We really appreciate it. Attached is the reply to your comments. 

Regards, 

Suzana 

Reviewer 2 Report

Thanks to the authors for diligent editing.

Author Response

Dear Reviewer, 

Thank you for your constructive feedback. We really appreciate it.

Regards, 

Suzana